# Human Milk Antibodies against S1 and S2 Subunits from SARS-CoV-2, HCoV-OC43, and HCoV-229E in Mothers with a Confirmed COVID-19 PCR, Viral SYMPTOMS, and Unexposed Mothers

**DOI:** 10.3390/ijms22041749

**Published:** 2021-02-09

**Authors:** Veronique Demers-Mathieu, Ciera DaPra, Gabrielle Mathijssen, David A. Sela, Kirsi M. Järvinen, Antti Seppo, Shawn Fels, Elena Medo

**Affiliations:** 1Department of Neonatal Immunology and Microbiology, Medolac Laboratories A Public Benefit Corporation, Boulder City, NV 89005, USA; dapra@unlv.nevada.edu (C.D.); mathijss@unlv.nevada.edu (G.M.); sfels@medolac.com (S.F.); emedo@medolac.com (E.M.); 2Department of Food Science, University of Massachusetts Amherst, Amherst, MA 01003, USA; davidsela@umass.edu; 3Division of Pediatric Allergy and Immunology & Center for Food Allergy, School of Medicine and Dentistry, University of Rochester, Rochester, NY 14642, USA; Kirsi_Jarvinen-seppo@urmc.rochester.edu (K.M.J.); Antti_Seppo@urmc.rochester.edu (A.S.)

**Keywords:** breast milk, passive immunity, immunoglobulins, infectious diseases, breastfeeding, polyreactive antibody, β-coronaviruses, α-coronaviruses, neonates

## Abstract

Background: Preexisting immunity to SARS-CoV-2 could be related to cross-reactive antibodies to common human-coronaviruses (HCoVs). This study aimed to evaluate whether human milk antibodies against to S1 and S2 subunits SARS-CoV-2 are cross-reactive to S1 and S2 subunits HCoV-OC43 and HCoV-229E in mothers with a confirmed COVID-19 PCR test, in mothers with previous viral symptoms during COVID-19 pandemic, and in unexposed mothers; Methods: The levels of secretory IgA (SIgA)/IgA, secretory IgM (SIgM)/IgM, and IgG specific to S1 and S2 SARS-CoV-2, and reactive to S1 + S2 HCoV-OC43, and HCoV-229E were measured in milk from 7 mothers with a confirmed COVID-19 PCR test, 20 mothers with viral symptoms, and unexposed mothers (6 Ctl1-2018 and 16 Ctl2-2018) using ELISA; Results: The S2 SARS-CoV-2 IgG levels were higher in the COVID-19 PCR (*p* = 0.014) and viral symptom (*p* = 0.040) groups than in the Ctl1-2018 group. We detected a higher number of positive correlations between the antigens and secretory antibodies in the COVID-19 PCR group than in the viral symptom and Ctl-2018 groups. S1 + S2 HCoV-OC43-reactive IgG was higher in the COVID-19 group than in the control group (*p* = 0.002) but did not differ for the other antibodies; Conclusions: Mothers with a confirmed COVID-19 PCR and mothers with previous viral symptoms had preexisting human milk antibodies against S2 subunit SARS-CoV-2. Human milk IgG were more specific to S2 subunit SARS-CoV-2 than other antibodies, whereas SIgA and SIgM were polyreactive and cross-reactive to S1 or S2 subunit SARS-CoV-2.

## 1. Introduction

Human milk antibodies possess several advantages to protect against COVID-19, including a high concentration of secretory IgA (SIgA) and the presence of secretory IgM (SIgM) [1,2,3]. SIgA is the most abundant antibody that protects mucosal surfaces against infectious microbes [4,5]. SIgM also possesses a superior capacity to bind antigens than IgM [6]. The secretory component is a critical constituent of SIgA and SIgM to perform immune exclusion and prevent pathogens’ invasion by blocking their attachment to the epithelial cells [7,8]. Human milk SIgA and SIgM are resistant to digestive proteases and low pH [2,6,9], allowing them to survive during infant digestion [1,3,9,10]. Most importantly, human milk secretory antibodies specific to SARS-CoV-2 could help to protect infants against COVID-19 infection. Clinical manifestations of pediatric COVID-19 have been related to the respiratory and gastrointestinal (GI) systems [11]. It is well-known that breastfeeding reduces the risk of viral infections in exclusively breastfed infants, but the mechanism of how human milk antibodies protect the respiratory tract against viral pathogens is still unclear [12].

The spike protein (S) is present on coronaviruses’ surface and is composed of two subunits, S1 and S2, that are essential for both host specificity and viral infectivity [13]. S1 subunit contains the receptor-binding domain (RBD) that recognizes the cell surface receptor [14]. S2 subunit contains the required components for membrane fusion [15]. Antibodies specific to S1 could disable the receptor interactions and block the viral infection process [13]. S1- and S2-specific antibodies could also destabilize the prefusion structure and block coronaviruses’ entry [13].

A recent study demonstrated that mothers with confirmed PCR SARS-CoV-2 had higher levels of antibodies specific to RBD SARS-CoV-2 than in unexposed mothers [16]. Dong et al. [17] showed that nucleocapsid-specific IgG and IgA levels were higher after five weeks of post-infection in women with confirmed PCR test than after two weeks. The difference of levels in human milk antibodies specific to S1 or S2 subunit SARS-CoV-2 in mothers with confirmed COVID-19 PCR test remains unknown.

Our recent study demonstrated that S1 + S2-reactive SARS-CoV-2 SIgA/IgA and SIgM/IgM were detected in most human milk samples collected during COVID-19, but mothers did not have a confirmed PCR test [18]. These mothers had viral symptoms associated with COVID-19 infection. We speculated that human milk secretory antibodies might recognize a broad range of human coronaviruses (HCoVs) due to their polyreactive and cross-reactive properties, but no study has evaluated this hypothesis.

HCoV-OC43 and HCoV-229E cause respiratory infections, such as acute respiratory tract disease and pneumonia. The incidence of HCoV-OC43 infection in hospitalized infants (from birth to six months) is more often than HCoV-HK1, HCoV-229E, and HCoV-NL63 infections (i.e., HCoV-OC43 > HCoV-NL63 > HCoV-HKU1 > HCoV-229E) [19]. Infants infected by HCoV-OC43 had immunity against HCoV-HKU1 infection [19]. This result suggests that cross-reactive antibodies to HCoV-OC43 may protect against infection by other HCoVs. The positive correlation between human milk antibodies reactive to S1 + S2 subunits from HCoV-OC43, HCoV-229E, and SARS-CoV-2 is still unexplored.

This study aimed to evaluate the levels of SIgA/IgA, SIgM/IgM, and IgG specific to SARS-CoV-2 S1 or S2 subunit and reactive to HCoV-OC43 and HCoV-229E S1 + S2 subunits in milk from seven women with positive COVID-19 PCR test and six unexposed women (Ctl1-2018 pre-pandemic) as well as between 20 mothers that had viral symptoms during COVID-19 pandemic (no PCR test) and 16 unexposed mothers (Ctl2-2018 pre-pandemic). These results could help identify the critical factors influencing the maternal antibody response. This investigation’s clinical relevance is that women with COVID-19 PCR test and those with previous viral symptoms (but no PCR test) could secrete significant levels of antibodies against SARS-CoV-2 in human milk and provide passive immunity to their infants against coronavirus infection.

## 2. Results

### 2.1. Maternal Demographics

Postpartum time, infant gender, and maternal age were comparable between the COVID-19 PCR and control 1 (Ctl1-2018) groups (Table 1) as well as between the viral symptom and control 2 groups (Ctl2-2018) (Table 2). Individual participant characteristics are described in Appendix A.

### 2.2. S1 or S2 Subunit SARS-CoV-2-Specific Human Milk Antibodies

The S2 SARS-CoV-2-specific IgG level was 2.8-fold higher in the COVID-19 group than in the Ctl1-2018 group (*p* = 0.014, Figure 1F) but did not differ for S1 SARS-CoV-2-specific IgG (Figure 1C). The levels of S1 or S2 SARS-CoV-2-specific SIgA/IgA and SIgM/IgM did not differ between COVID-19 and Ctl1-2018 groups (Figure 1A,B,D,E). Mother-1 (M1) had the highest levels of S1 or S2 antibodies among the donors. Among the control group, C4 and C5 had high levels of S1 or S2 SARS-CoV-2 IgG and SIgM/IgM.

### 2.3. S1- and S2 Subunits HCoVs-Reactive Human Milk Antibodies

S1 + S2 HCoV-OC43 IgG was 4.3-fold higher in the COVID-19 group than in the Ctl1-2018 group (*p* = 0.002, Figure 2C), but S1 + S2 HCoV-229E IgG did not differ (Figure 2F). SIgM/IgM and SIgA/IgA reactive to HCoV-OC43 and HCoV-229E S1 + S2 did not differ between COVID-19 and Clt1-2018 groups (Figure 2A,B,D,E). M1 had the highest SIgA/IgA and SIgM/IgM reactive to S1 + S2 from HCoV-OC43 and HCoV-229E.

### 2.4. S1 or S2 Subunit SARS-CoV-2-Reactive Human Milk Antibodies

The S2 SARS-CoV-2-reactive IgG level was higher in the viral symptom group than in the Ctl2-2018 group (Figure 3E,F) but did not differ for the other antibodies. Mother-5 (m5) had the most frequent elevated antibodies reactive to SARS-CoV-2 among the donors, followed by mother-17 (m17), mother-20 (m20), and mother-9 (m9). Among the Ctl2-2018 group, c7 had the highest level of SARS-CoV-2 SIgA/IgA and SIgM/IgM.

### 2.5. S1- and S2 Subunits HCoVs-Reactive Human Milk Antibodies

S1 + S2 HCoV-OC43 and HCoV-229E antibodies did not differ between the viral symptom group than in the Ctl2-2018 group (Figure 4A–F).

### 2.6. Correlation Matrix between Antigens and Isotypes

#### 2.6.1. S1 or S2 Subunit from SARS-CoV-2

In the COVID-19 group, 26 positive correlations were detected between S1 and S2 from SARS-CoV-2 for SIgA/IgA and SIgM/IgM (Figure 5A). S1 SARS-CoV-2 IgG was positively correlated with S1 or S2 SARS-CoV-2 SIgA/IgA and SIgM/IgM, but no correlation was detected for S2 SARS-CoV-2 IgG.

In the viral symptom group, we detected positive correlations between S1 SARS-CoV-2 SIgA/IgA and S2 SARS-CoV-2 SIgA/IgA, between S1 SARS-CoV-2 SIgM/IgM and S2 SARS-CoV-2 SIgM/IgM, and between S1 SARS-CoV-2 IgG and S2 SARS-CoV-2 IgG (Figure 5B).

In the Ctl-2018 groups (Ctl1 + Ctl2), S1 SARS-CoV-2 SIgA/IgA was positively correlated with S2 SARS-CoV-2 SIgA/IgA and SIgM/IgM. S2 SARS-CoV-2 SIgM/IgM was positively correlated with S1 SARS-CoV-2 SIgM/IgM, and IgG, and S2 SARS-CoV-2 SIgA/IgA (Figure 5C).

#### 2.6.2. S1 and S2 Subunits from SARS-CoV-2 and HCoVs

In the COVID-19 group, S1 and S2 SARS-CoV-2 SIgA/IgA positively correlated with HCoV-OC43 SIgA/IgA and SIgM/IgM, HCoV-229E SIgA/IgA, and SIgM/IgM, but did not correlate with HCoVs IgG (Figure 5). S1 and S2 SARS-CoV-2 SIgM/IgM positively correlated with HCoVs SIgM/IgM but did not correlate with HCoVs IgG and SIgA/IgA. No correlation was detected for S1 + S2 HCoVs IgG. HCoV-OC43 SIgA/IgA was positively correlated with HCoV-229E SIgA/IgA and SIgM/IgM. A positive correlation was observed between HCoV-OC43 SIgM/IgM and HCoV-229E SIgM/IgM.

In the viral symptom group, S1 + S2 HCoV-229E SIgM/IgM was positively correlated with S1 SARS-CoV-2 IgG and S2 SARS-CoV-2 SIgM/IgM. S1 + S2 HCoV-229E IgG was positively correlated with S2 SARS-CoV-2 IgG, S1 + S2 HCoV-OC43-IgG, S1 + S2 HCoV-229E SIgA/IgA (Figure 5B).

In the Ctl-2018 group, positive correlations were detected between S1 + S2 HCoV-OC43 IgG and SIgM/IgM and between S1 + S2 HCoV-229E SIgA/IgA and S1 + S2 HCoV-OC43.

The time between the infection and milk collection did not correlate with antibodies’ levels to SARS-CoV-2, HCoV-OC43, and HCoV-229E (Figure 5C). The differences in amino acid sequences between the S1 subunit and S2 subunit from SARS-CoV-2 and S1 + S2 subunits from HCoV-229E and HCoV-OC43 are illustrated in Appendix A. The concentration in SIgA/IgA, SIgM/IgM, and IgG did not differ between COVID-19 PCR, viral symptom, and Ctl-2018 groups (overall means ± SD: 1427 ± 521 μg/mL for SIgA/IgA, 3 ± 2 μg/mL for SIgM/IgM, and 10 ± 6 μg/mL for IgG).

## 3. Discussion

Infants and children (<2 years old) cannot receive the SARS-CoV-2 vaccine due to their immature antibody response [20]. While respiratory symptoms are the main clinical manifestations of COVID-19, gastrointestinal symptoms are also reported in patients (including infants and children) infected with SARS-CoV-2 [21]. Therefore, human milk antibodies specific to SARS-CoV-2 could protect the infant’s gut against COVID-19 infection. The potential protective effects of human milk antibodies may also decrease viral infection risk by other HCoVs. Antibodies that bind to S1 or S2 subunit could block the attachment or fusion of SARS-CoV-2. Our recent study demonstrated human milk antibodies’ presence reactive to SARS-CoV-2 S1 + S2 subunits in milk from mothers during the COVID-19 pandemic [18]. However, the levels in antibodies specific to S1 and S2 subunits from SARS-CoV-2 in mothers with confirmed COVID-19 PCR remains to be evaluated. This present study compared the levels of antibodies against S1 or S2 subunit SARS-CoV-2, S1 + S2 subunits HCoV-OC43, and HCoV-229E between human milk collected from mothers diagnosed COVID-19 via PCR, from mothers with previous viral symptoms during COVID-19 pandemic, and from unexposed mothers in 2018. The study’s clinical relevance was to determine whether COVID-19 PCR mothers and mothers with previous viral symptoms had preexisting human milk antibodies against SARS-CoV-2 partially due to cross-reactive antibody reactive to HCoV-OC43 and HCoV-229E.

For the first time, we demonstrated that S2 subunit SARS-CoV-2-specific IgG levels were higher in the COVID-19 PCR and viral symptom groups than in the Ctl-2018 group, but the other antibodies did not differ. While S2 is less exposed than the S1 subunit on SARS-CoV-2, IgG may recognize S2 during the viral entry’s complex conformation change [14,22]. Recovered COVID-19 individuals with cross-reactive B cell responses against the S2 subunit may enhance broad coronavirus protection [23]. Antibodies targeting the S2 subunit had neutralizing activity, suggesting that the presence of S2-reactive IgG provided some protection against SARS-CoV-2 [24]. Another study [25] found that the epitope S2-78 exhibited potent neutralizing activity by interfering with the formation of 6-HB (helical bundle), an essential structure for cell membrane fusion. Most studies focus on the RBD region, which may induce potential mutations in this region and reduce the effectiveness of the RBD-specific therapeutic antibodies and vaccines [26]. The identification of other targets that can confer neutralizing antibodies is critical to overcoming future SARS-CoV-2 mutations.

The lack of difference between COVID-19 PCR, viral symptom, and Ctl-2018 groups for SARS-CoV-2-reactive SIgA/IgA and SIgM/IgM and the high numbers of positive correlations between antigen and secretory antibodies in the COVID-19 PCR group could be related to their polyreactive capacity to bind different epitopes [27,28]. These findings are in accordance with our recent study [18], where S1 + S2 subunits SARS-CoV-2 IgG was higher in milk from women with viral symptoms (but no PCR test) during the COVID-19 pandemic 2020 than in the control group 2018. Moreover, the correlation matrix generated with the Ctl-2018 group had a significantly smaller number of positive correlations than those generated with the COVID-19 group. This observation suggests that human milk antibodies were more specific to SARS-CoV-2 in the COVID-19 PCR group than in the Ctl-2018 group.

The highest levels of antibodies specific to S1 or S2 subunit SARS-CoV-2 was observed in mother-1 with a confirmed + PCR test (M1). This observation could be related to the highest level of long-lived specific T cells that enhance B cell antibody production among COVID-19 mothers. M1 could also have antibodies that were disappearing less quickly (longer half-life) than the other mothers due to their immune status or viral exposure route. The time between the infection and collection did not affect the levels of antibodies specific to S1 or S2 SARS-CoV-2. Individual mothers have different levels of human milk antibody probably due to the differences of previous infections, preexisting immunity, age, genetic factors, and other factors affecting the immune response [29,30].

S1 + S2 subunits HCoV-OC43-reactive IgG was higher in the COVID-19 group than in the control group but did not differ for S1 + S2 subunits HCoV-229E-reactive IgG or other antibodies. These findings are consistent with a recent study [23]. IgG levels against S protein of HCoV-OC43 (but not against HCoV-229E) were higher in convalescent subjects than in non-SARS-CoC-2-exposed subjects and correlated strongly with anti-S2 IgG levels [23]. B cells may have a stronger cross-reactivity between the S2 subunits of SARS-CoV-2 and human β-coronaviruses than α-coronaviruses [31]. Moreover, we observed high SIgM/IgM levels reactive to SARS-CoV-2 S1 subunit and S1 + S2 subunits HCoV-OC43 and HCoV-229E in two unexposed mothers (Ctl1-2018) C4) and C5. These observations suggest the presence of cross-reactive antibodies against human coronaviruses. SARS-CoV-2, HCoV-OC43, and HCoV-229E could share similar epitopes on S1 and S2 subunits that are recognized by human milk antibodies. Our findings are in agreement with recent articles suggesting immunity to “common cold” coronaviruses [32] may produce cross-reactive antibodies to SARS-CoV-2. Grifoni et al. [33] detected SARS-CoV-2-reactive CD4^+^ T cells in ~40–60% of unexposed individuals (blood), suggesting cross-reactive T cell recognition between circulating “common cold” coronaviruses and SARS-CoV-2. SARS-CoV-2 RBD-reactive IgG, IgM, and IgA in serums from non-exposed individuals were detected, but the serum antibody titer was higher in individuals with COVID-19 than in serum from unexposed individuals [33]. Mateus et al. [32] demonstrated a range of preexisting memory CD4^+^ T cells that were cross-reactive to SARS-CoV-2, HCoV-OC43, HCoV-229E, HCoV-NL63, and HCoV-HKU1. S-reactive T cells in patients with COVID-19 responded similarly to HCoV-229E S and SARS-CoV-2 S, suggesting S-cross-reactive T cells’ presence was probably generated during past encounters with endemic coronaviruses [31]. Preexisting SARS-CoV-2-reactive antibodies could be relevant because mothers with a high-level of preexisting antibodies recognizing SARS-CoV-2 could provide stronger passive immunity to their infants and reduce the risk of COVID-19 and common cold coronaviruses infections.

During COVID-19 infection in lactating women, SARS-CoV-2-specific IgA and IgM could be produced by plasma cells in the interstitial fluid of the mammary gland tissues (MEC) and transported by polymeric immunoglobulin receptor (PIgR) across the MEC (Figure 6A–G) [9]. SIgA and SIgM are in the highest concentration than IgA or IgM due to this diffusion mechanism in the alveolar lumen. However, there is some proportion of IgA and IgM (without SC) in human milk, likely due to the production of antibodies by B cells present in human milk [1]. SARS-CoV-2-reactive IgG could be produced in the maternal blood after contact with SARS-CoV-2 or other coronaviruses. To access the alveolar lumen, IgG needs to bind to the neonatal Fc receptor (FcRn) on the basolateral membrane of the MEC (Figure 6H–K) [34]. Deactivated or intact viruses may activate the adaptive immunity to recognize the viral proteins from coronaviruses, including S1 subunit, S2 subunit, RBD, and nucleocapsid protein (Figure 6L). As spike proteins from SARS-CoV-2, HCoV-OC43, and HCoV-229E share structural similarities, antibodies produced by B cells after antigen recognition could be cross-reactive to these human coronaviruses. Human milk antibodies against SARS-CoV-2 may provide additional immune defense to infants and reduce the risk of COVID-19 infection, but their neutralizing capacity remains to be investigated.

There are a few limitations in this study. First, the neutralizing capacity of HM antibodies against SARS-CoV-2 was not determined. Second, this study had a small sample size, but our sample size proved to be adequately powered based on the results.

## 4. Materials and Methods

### 4.1. Study Design and Participants

For the first experiment, a screening survey was completed to recruit donors that had a confirmed COVID-19 PCR test. Participants reported when they had the positive PCR test, the duration, and their symptoms. Milk samples for the first control group (Ctl1-2018) were collected from mothers in the 2018 pre-pandemic COVID-19 to match the demographic characteristics of the COVID-19 group. For the second experiment, a screening survey was completed to recruit donors that had previous viral symptoms during the COVID-19 pandemic but were not able to confirm the SARS-CoV-2 infection by PCR. Participants reported when they were sick and their symptoms. The inclusion criteria were living in the USA, lactation time between 4 and 10 months, passing blood tests, and completing a health questionnaire. An additional criterion for the mothers in the COVID-19 PCR group was a positive PCR test. Written consents to use their milk for research were obtained from all participants. The blood test was performed to exclude women infected with HIV, hepatitis C virus, hepatitis B virus, or syphilis, to reduce the risk of cross-reactive antibodies between viruses. These viral infectious diseases could change the maternal immune system (impaired immune response and antibody production [35]), thus influencing the results. Lactation time between 4 and 10 months was selected to reduce the effect of lactation time on the levels of antibodies. Our previous studies [30,36] demonstrated that antibody concentrations were stable between 4 to 10 months of lactation. Donors were approved through Mothers Milk Cooperative. Milk collection was approved by the institutional review board (IRB00012424) of Medolac Laboratories. The exclusion criteria were mothers who were smoking, taking medications, and drugs.

### 4.2. Human Milk Collection

Human milk samples (150–250 mL) were collected at home with clean electric breast pumps into sterile plastic containers and stored immediately at −20 °C in deep freezers. The breast was cleaned with a wet washcloth (no soap or alcohol) before pumping. HM samples were frozen and transported in insulated boxes to Medolac Laboratories and stored at −30 °C until the ELISA measurements.

### 4.3. Human Milk Antibody Detection

The levels of SIgM/IgM, SIgA/IgA, and IgG reactive to S1 or S2 subunit SARS-CoV-2 and S1 + S2 subunits from HCoV-OC43 and HCoV-229E were determined using ELISAs. The protocols used are described in our recent study with some modifications [18]. Briefly, ELISAs were recorded with a microplate reader (SpectraMax iD5, Molecular Devices, Sunnyvale, CA, USA). Clear, flat-bottomed microplates (Nunc MaxiSorp, Thermo Fisher Scientific, Rochester, NY) were coated with 100 µL of recombinant S1 or S2 subunit from SARS-CoV-2 (2019-nCoV) or recombinant S1 + S2 subunits from HCoV-OC43 or HCoV-229E (Sino Biological US Inc., Wayne, PA, USA) at 1 µg/mL in 1x phosphate-buffered saline pH 7.4 (PBS, Gibco, Grand Island, NY, USA). These antigens were expressed by baculovirus-insect cells with a polyhistidine tag at the C-terminus from DNA sequences encoding SARS-CoV-2 S1 or S2 subunit (YP_009724390.1) (Met1-Arg685 for S1 and Ser686-Pro1213 for S2), HCoV-OC43 S1 + S2 subunits (AVR40344.1) (Met1-Pro1304), HCoV-229E S1 + S2 subunits (APT69883.1) (Cyst16-Trp1115). Microplates were incubated overnight at 4 °C. After incubation, plates were washed 3 times using PBS with 0.05% Tween-20 detergent (PBST) (Thermo Fisher Scientific, Rochester, NY, USA) and then 200 µL of blocking buffer (PBST with 3% of bovine serum albumin fraction V (Roche Diagnostic GmbH, Manheim, Germany) was added in all wells for 1 h at room temperature. Frozen human milk samples were thawed and centrifuged (1301× *g* for 20 min at 4 °C) to collect the supernatants. Standard samples were prepared using human milk supernatants with the highest optical density (OD) value for each ELISA. Different standards were used due to the difference in OD values between the different isotypes and the antigens. These standards were selected based on the preliminary data of supernatants diluted at 10× for IgG and SIgM/IgM and 25x for SIgA/IgA (Appendix A). The levels of antibodies were derived by interpolation from the standard curves and assigned quantity expressed in arbitrary units/mL (U/mL) (Appendix A). For each step (addition of 100 µL standards/samples and secondary antibodies at 1 µg/mL), washing and incubation for 1 h at room temperature were performed. The detection was completed using goat anti-human IgM mu-chain HRP for SIgM/IgM, goat anti-human gamma-chain HRP for IgG, and goat anti-human alpha-chain HRP for SIgA/IgA (Abcam, Cambridge, MA, USA). Concentrations of SIgA/IgA, SIgM/IgM, and IgG were determined using ELISA as described in our previous study [36].

### 4.4. Statistical Analysis

Mann–Whitney tests (unpaired experimental design) were applied using GraphPad Prism (version 8) to compare human milk antibodies’ levels between COVID-19 PCR and the Ctl1-2018 groups as well as between viral symptom and Ctl2-2018 groups. Three matrix correlations (one for COVID-19 PCR, one for viral symptoms, and another for two control-2018) were performed between variables used to determine correlations between the isotypes and antigens. Linear regressions were evaluated between the infection’s time to the milk collection and the antibody levels in the COVID-19 PCR group. The sample size was selected based on our previous studies of sample sizes [3,18] and proved to be adequately powered based on the results.

## 5. Conclusions

In summary, this study demonstrates that cross-reactive human milk antibodies are present at varying levels in COVID-19 PCR, viral symptom, and pre-pandemic groups. The level of S2 subunit SARS-CoV-2 IgG was higher in the COVID-19 PCR and viral symptom groups than in the Ctl-2018 groups. Mothers with a confirmed COVID-19 PCR or previous viral symptoms may produce B cell response cross-reactive to the S2 subunit that promotes broad protection to their breastfed infants against infection from SARS-CoV-2 mutations. The high numbers of positive correlations between antigens and secretory antibodies (SIgA and SIgM) in the COVID-19 PCR group and the absence of significant difference in SIgA or SIgM levels specific to S1 or S2 subunit SARS-CoV-2 between COVID-19 PCR and unexposed mothers could be related to their polyreactive capacity to bind different epitopes on the subunits S1 and S2. The difference in the neutralizing capacity of antibodies between COVID-19 PCR, viral symptom, and pre-pandemic groups needs to be investigated to draw a clear conclusion regarding the protective effect of human milk antibodies against SARS-CoV-2. S1 + S2-reactive HCoV-OC43 IgG was higher in the COVID-19 group than in the Ctl1-2018 group but did not differ for HCoV-229E, revealing a stronger cross-reactivity between the S2 subunits of SARS-CoV-2 and human β-coronaviruses than α-coronaviruses.

## Figures and Tables

**Figure 1 ijms-22-01749-f001:**
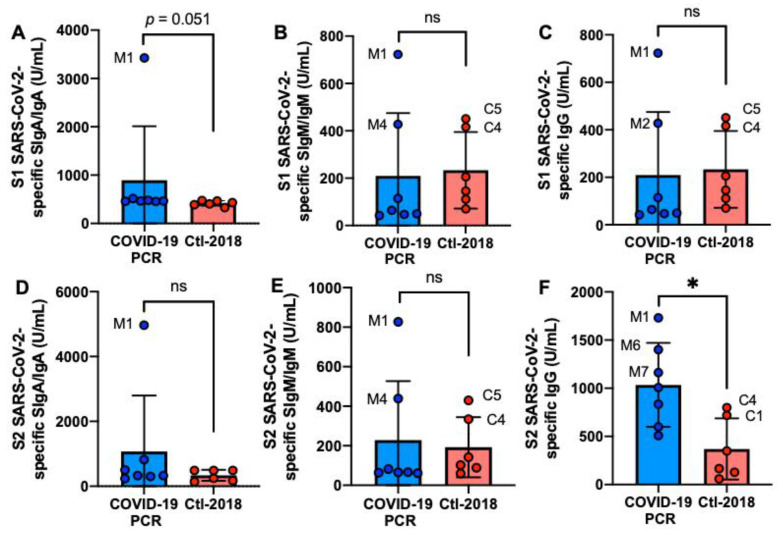
Levels of human milk antibodies specific to SARS-CoV-2 S1 or S2 subunit in mothers with confirmed COVID-19 PCR test and unexposed mothers. (**A**,**B**) secretory IgA (SIgA)/IgA; (**B**,**E**) secretory IgM (SIgM)/IgM; (**C**,**F**) IgG. (**A**–**C**) S1 SARS-CoV-2 antibodies; (**D**–**F**) S2 SARS-CoV-2 antibodies. Values are means ± SD, *n* = 7 for mothers with confirmed COVID-19 PCR test, and *n* = 6 for unexposed mothers (Clt1-2018 pre-pandemic). Mann–Whitney test was used to compare the two groups. Asterisk shows statistically significant differences between variables (* *p* < 0.05). ns, not significant.

**Figure 2 ijms-22-01749-f002:**
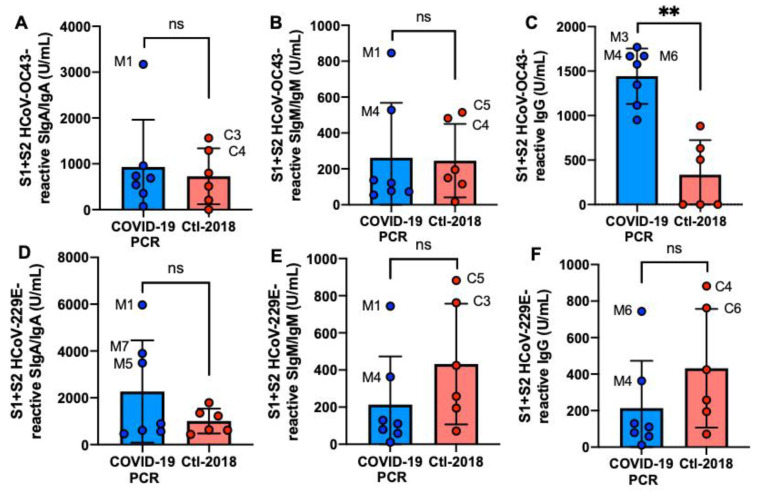
Levels of human milk antibodies reactive to HCoV-OC43 and HCoV-229E S1 + S2 subunits between mothers with confirmed COVID-19 PCR test and unexposed mothers. (**A**,**D**) secretory IgA (SIgA)/IgA; (**B**,**E**) secretory IgM (SIgM)/IgM; (**C**,**F**) IgG. (**A**–**C**) S1 + S2 HCoV-OC43-reactive antibodies; (**D**–**F**) S1 + S2 HCoV-229E-reactive antibodies. Values are means ± SD, *n* = 7 for mothers with confirmed COVID-19 PCR test and *n* = 6 for unexposed mothers (Clt1-2018 pre-pandemic). Mann–Whitney test was used to compare the two groups. The asterisks show statistically significant differences between variables (** *p* < 0.01). ns, not significant.

**Figure 3 ijms-22-01749-f003:**
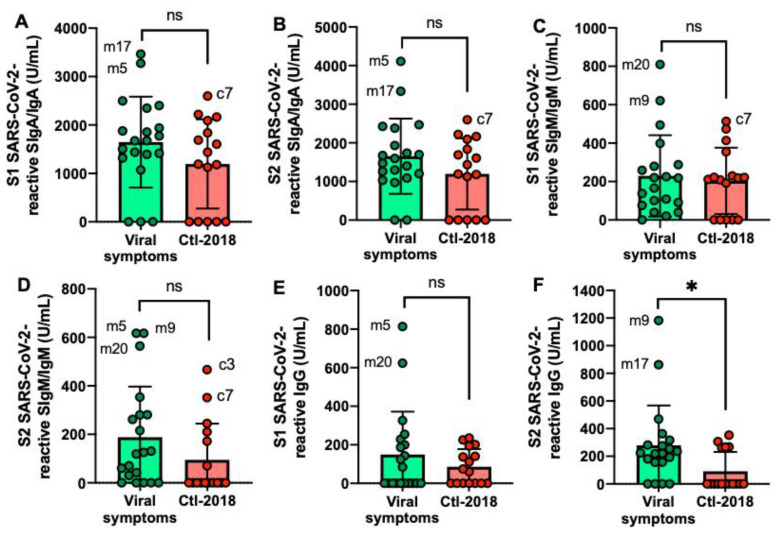
Levels of human milk antibodies specific to SARS-CoV-2 S1 or S2 subunit in mothers with previous viral symptoms during COVID-19 pandemic (no PCR test) and unexposed mothers. (**A**,**B**) secretory IgA (SIgA)/IgA; (**B**,**E**) secretory IgM (SIgM)/IgM; (**C**,**F**) IgG. (**A**–**C**) S1 SARS-CoV-2 antibodies; (**D**,**F**) S2 SARS-CoV-2 antibodies. Values are means ± SD, *n* = 20 for mothers with viral symptoms, and *n* = 16 for unexposed mothers (Clt2-2018 pre-pandemic). Mann–Whitney test was used to compare the two groups. Asterisk shows statistically significant differences between variables (* *p* < 0.05). ns, not significant.

**Figure 4 ijms-22-01749-f004:**
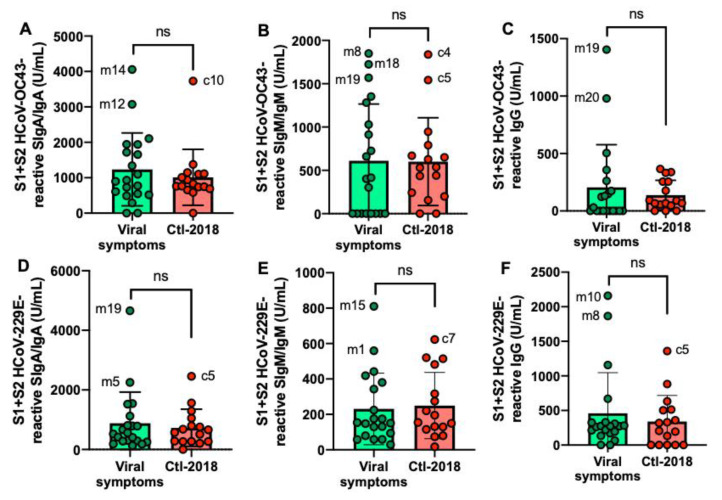
Levels of human milk antibodies reactive to HCoV-OC43 and HCoV-229E S1 + S2 subunits between mothers with previous viral symptoms during COVID-19 pandemic (no PCR test) and unexposed mothers. (**A**,**D**) secretory IgA (SIgA)/IgA; B, E secretory IgM (SIgM)/IgM; (**C**,**F**) IgG. (**A**–**C**) S1 + S2 HCoV-OC43-reactive antibodies; (**D**–**F**) S1 + S2 HCoV-229E-reactive antibodies. Values are means ± SD, *n* = 20 for mothers with viral symptoms and *n* = 16 for unexposed mothers (Clt1-2018 pre-pandemic). Mann–Whitney test was used to compare the two groups. ns, not significant.

**Figure 5 ijms-22-01749-f005:**
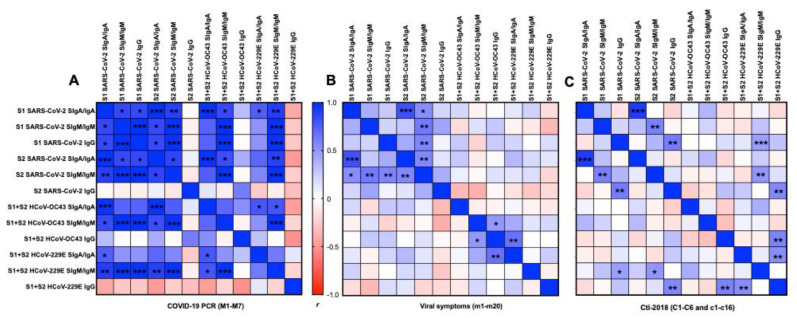
Matrix correlations between the type of antigens (S1 or S2 subunit SARS-CoV-2, S1 + S2 HCoV-OC43, and HCoV-229E) and the type of isotypes (secretory SIgA (SIgA)/IgA, secretory SIgM (SIgM)/IgM, or IgG) in human milk. (**A**) Mothers with confirmed COVID-19 PCR (*n* = 7) and (**B**) mothers with previous viral symptoms during COVID-19 pandemic (*n* = 20), and (**C**) unexposed mothers (*n* = 22, for Ctl1-2018 (*n* = 6) and for Ctl2-2018 (*n* = 16), pre-pandemic). Positive correlations are in blue and negative correlations are in red for Pearson’s correlation coefficients values (*r*). Asterisks show statistically significant differences between variables (*** *p* < 0.001, ** *p* < 0.01, and * *p* < 0.05) using matrix correlation (two-tailed and 95% of confidence interval).

**Figure 6 ijms-22-01749-f006:**
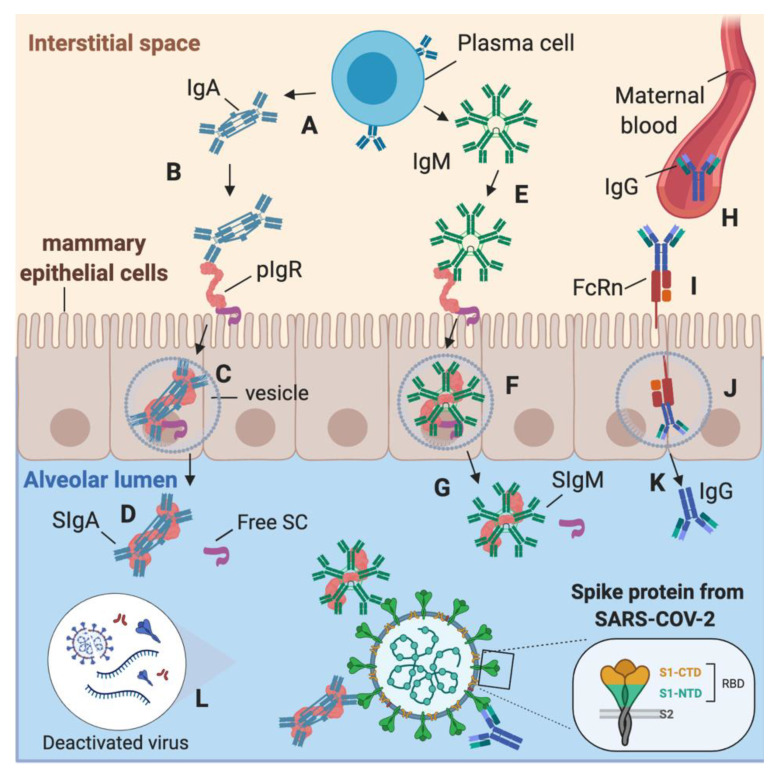
Origin of secretory IgA (SIgA), secretory IgM (SIgM) and IgG antibodies specific to SARS-CoV-2 in human milk. (**A**) In the mammary gland tissues’ interstitial space, plasma cell secretes IgA, which can bind to the polymeric Ig receptor (pIgR). (**B**) IgA binds via their Fc region to pIgR to form the pIgR–IgA complex. (**C**) The pIgR–IgA is transported in a vesicle across the mammary epithelial cells (MEC) to the alveolar lumen. (**D**) SIgA is released from the pIgR–IgA complex by proteolytic cleavage into the alveolar lumen, and then SIgA is diffused in human milk. (**E**) IgM is produced by plasma cells in the interstitial space and can bind to pIgR to form the complex pIgR–IgM. (**F**) pIgR–IgM is transported in a vesicle across the MEC to be released in the alveolar lumen. (**G**) SIgM is released from the pIgR–IgM complex by proteolytic cleavage into the alveolar lumen, and then SIgM is diffused in human milk. (**H**) IgG can diffuse from maternal blood after binding to neonatal Fc receptor (FcRn) (**I**) on the basolateral membrane of the MEC. (**J**) FcRn transport IgG via a vesicle to (**K**) the alveolar lumen. (**L**) Deactivated or intact viruses in human milk could activate the adaptive immunity to recognize the viral proteins from coronaviruses, including S1, S2, RBD, and nucleocapsid. Created with BioRender.com.

**Table 1 ijms-22-01749-t001:** Demographic description (self-reported) of mothers with confirmed COVID-19 PCR test and unexposed mothers (Ctl1-2018 pre-pandemic).

Demographics	COVID-19 PCR (*n* = 7) ^b^	Ctl1-2018 (*n* = 6)
Postpartum time, months ^a^	6 ± 1 (5–8)	6 ± 1 (5–8)
Infant gender, *n*	4 males: 3 females	3 males: 3 females
Maternal age, years ^a^	31 ± 4 (26–37)	31 ± 4 (27–37)
Influenza vaccine during pregnancy, *n* (%)	2 (28.6)	5 (83.3)
Time from infection to collection, days	47 ± 24 (16–84)	-

^a^ Data are means ± SD, min, and max. ^b^ Women were diagnosed with COVID-19 PCR test by a nasal swab (positive RNA SARS-CoV-2) between May 2020 and November 2020. The viral symptoms reported were cough, fatigue, fever, sore throat, headaches, and loss of taste/smell. Milk collection was performed after the COVID-19 infection.

**Table 2 ijms-22-01749-t002:** Demographic description (self-reported) of mothers with previous viral symptoms during the COVID-19 pandemic (viral symptoms) and unexposed mothers (Ctl2-2018 pre-pandemic).

Demographics	Viral Symptoms (*n* = 20) ^b^	Ctl2-2018 (*n* = 16)
Postpartum time, months ^a^	5 ± 2 (2–11)	5 ± 3 (5–8)
Infant gender, *n*	8 males: 12 females	8 males: 8 females
Maternal age, years ^a^	32 ± 4 (27–41)	32 ± 4 (25–37)
Influenza vaccine during pregnancy, *n* (%)	7 (35)	6 (37.6)

^a^ Data are means ± SD, min, and max. ^b^ Women had symptoms of viral respiratory infection (cold, fever, respiratory infection, sinus infection, nasal congestion, headaches, sore throat and/or flu-like symptoms) between April 2020 to September 2020, but no infection during the milk collection. The time between the infection and collection is unknown.

## Data Availability

Data sharing not applicable.

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
