# Peer review of "Human Milk Antibodies against S1 and S2 Subunits from SARS-CoV-2, HCoV-OC43, and HCoV-229E in Mothers with a Confirmed COVID-19 PCR, Viral SYMPTOMS, and Unexposed Mothers"

_ijms, 2021, doi:10.3390/ijms22041749_

Round 1
Reviewer 1 Report
Dear authors!
I read the manuscript you submitted to IJMS with high interest.
The comparison of antibodies from HCoV-43 and HCoV-229 to COVID-19 ones is of high interest. I would like to recommend you add some details to the paper.
1) In the literature, S1 and S2 are named as proteins, as well as subunits. In the context of the results you have obtained, I suggest using "S1 subunit" and "S2 subunit". Since the use of "polyreactive" and "cross-reactive" antibodies to the same protein (S1 and S2 subunits) can mess a reader.
2) The sequence of S1 and S2 proteins of COVID-19 and HCoV-43, and HCov-229 seems to be known. Please add a figure illustrating the differences of the amino acid sequences (maybe in Supplementary).
3) Please add the concentrations of IgG, sIgM, and sIgA in the donor's milk.
4) Please describe how Figs 1-4 and Fig 5 were created.
5) Please discuss the biological significance of milk antibodies against the different subunits (S1, S2) of COVID-19 S-protein.
6) Please describe how the S1 and S2 were synthesized by Sino Biological.
I'm sure that your paper will be interesting to many readers.
Sincerely
Author Response
Response to Reviewers
Manuscript ID: ijms-1093026
Type of manuscript: Article
Title: Human milk antibodies against S1 and S2 subunits from SARS-CoV-2, HCoV-OC43, and HCoV-229E in mothers with a confirmed COVID-19 PCR and unexposed mothers
We have made all the reviewer requested changes and hope you agree that the revised manuscript is now suitable for publication. Edited text in the revised manuscript is highlighted yellow. We have also responded to each of the reviewer comments below. We would like to take this opportunity to thank the reviewers for helping to improve our manuscript.
Reviewer 1
Dear authors!
I read the manuscript you submitted to IJMS with high interest.
The comparison of antibodies from HCoV-43 and HCoV-229 to COVID-19 ones is of high interest. I would like to recommend you add some details to the paper.
1) In the literature, S1 and S2 are named as proteins, as well as subunits. In the context of the results you have obtained, I suggest using "S1 subunit" and "S2 subunit". Since the use of "polyreactive" and "cross-reactive" antibodies to the same protein (S1 and S2 subunits) can mess a reader.
>> Thank you for this comment. We added “S1 subunit” and “S2 subunit” in the manuscript.
2) The sequence of S1 and S2 proteins of COVID-19 and HCoV-43, and HCov-229 seems to be known. Please add a figure illustrating the differences of the amino acid sequences (maybe in Supplementary).
>> Thank you. We added the Figure S1 to illustrate the difference in amino acid sequences between S1 subunit and S2 subunit from SARS-CoV-2 and S1+S2 subunits from HCoV-229E and HCoV-OC43. We referred the figure S1 in p. 7, line 192.
3) Please add the concentrations of IgG, sIgM, and sIgA in the donor's milk.
>> We added in the result that the concentration in SIgA/IgA, SIgM/IgM, and IgG did not differ between COVID-19 PCR, viral symptom, and Ctl-2018 groups (overall means ± SD: 1,427 ± 521 mg/mL for SIgA/IgA, 3 ± 2 mg/mL for SIgM/IgM, and 10 ± 6 mg/mL for IgG). (p. 7, lines 192-195. Concentrations of SIgA/IgA, SIgM/IgM, and IgG were determined using ELISA as de-scribed in our previous study. (p. 11, lines 377-378)
4) Please describe how Figs 1-4 and Fig 5 were created.
>> Thank you. We added Fig. S2, Fig. S3, and Fig. S4 to describe how the figure 1-5 were created using ELISA. We added to refer to these figures in the section methods (p. 11, lines 371-374)
5) Please discuss the biological significance of milk antibodies against the different subunits (S1, S2) of COVID-19 S-protein.
Thank you for this great comment. We added the biological significance of milk antibodies against S2 subunit of SARS-CoV-2 in the discussion. (p. 8, line 227-236 and lines 259-263)
6) Please describe how the S1 and S2 were synthesized by Sino Biological.
>> These antigens were expressed by Baculovirus-insect cells with a polyhistidine tag at the C-terminus from DNA sequences encoding SARS-CoV-2 S1 or S2 subunit (YP_009724390.1) (Met1-Arg685 for S1 and Ser686-Pro1213 for S2), HCoV-OC43 S1+S2 subunits (AVR40344.1) (Met1-Pro1304), HCoV-229E S1+S2 subunits (APT69883.1) (Cyst16-Trp1115). We added this information in the method section (p. 11, lines 357-361).
I'm sure that your paper will be interesting to many readers.
>> Thank you so much for your helpful comments. It really improved the quality and biological/clinical relevance of this manuscript.
Sincerely
Reviewer 2 Report
The manuscript by Demers-Mathieu et al. reports the antibody levels against SARS-CoV-2 and HCoVs in the breast milk of COVID-19-infected and pre-exposed mothers. The antibody levels were extensively determined and their correlations analyzed. I have a few minor points to be addressed before the manuscript could be considered for publication.
- In some occasions the authors failed to make clear distinction between the antigen and the antibody; for example, in lines 208-210, it is stated that " This present study compared the levels of S1 and S2 SARS-CoV-2, S1+S2 HCoV-OC43, and HCoV-229E between human milk collected from mothers diagnosed COVID-19 via PCR,..." when in fact the levels of antibodies against S1 and S2 were compared. The authors need to look for other such incidences throughout the manuscript.
- In Fig. 5 legends, (c) is missing.
- There are two Figure 5s. The second one should be Figure 6, along with its references in the main text.
- In page 6, line 182, S1_S2 should be S1+S2.
- A period is missing at the very end of the Discussion section (line 293)
- Line 241, check for the correct use of parentheses.
- Importantly, the authors need to provide more of their opinions on the observed antibody levels, their differences and correlations among the cohorts, subclass, and target antigens. For example, why is the anti-S2 IgG levels are significantly different between infected vs. uninfected individuals, but not the anti-S1 antibodies or different subclasses (IgA and IgM)? What is the plausible explanation for the observed correlations shown in Fig. 5? Can any conclusions/predictions could be drawn from such hypotheses, especially regarding the protective effect of breast milk against COVID-19?
Author Response
Response to Reviewers
Manuscript ID: ijms-1093026
Type of manuscript: Article
Title: Human milk antibodies against S1 and S2 subunits from SARS-CoV-2, HCoV-OC43, and HCoV-229E in mothers with a confirmed COVID-19 PCR and unexposed mothers
We have made all the reviewer requested changes and hope you agree that the revised manuscript is now suitable for publication. Edited text in the revised manuscript is highlighted yellow. We have also responded to each of the reviewer comments below. We would like to take this opportunity to thank the reviewers for helping to improve our manuscript.
Reviewer 2
The manuscript by Demers-Mathieu et al. reports the antibody levels against SARS-CoV-2 and HCoVs in the breast milk of COVID-19-infected and pre-exposed mothers. The antibody levels were extensively determined and their correlations analyzed. I have a few minor points to be addressed before the manuscript could be considered for publication.
- In some occasions the authors failed to make clear distinction between the antigen and the antibody; for example, in lines 208-210, it is stated that " This present study compared the levels of S1 and S2 SARS-CoV-2, S1+S2 HCoV-OC43, and HCoV-229E between human milk collected from mothers diagnosed COVID-19 via PCR,..." when in fact the levels of antibodies against S1 and S2 were compared. The authors need to look for other such incidences throughout the manuscript.
>> Thank you. We added “of antibodies against” in the sentence. (p. 7, line 216) We did not find other missing distinction between the antigen and the antibody throughout the manuscript.
- In Fig. 5 legends, (c) is missing.
>> Thank you. We added the missing “(c)” in the Fig. 5 legend (p. 7, line 200).
- There are two Figure 5s. The second one should be Figure 6, along with its references in the main text.
>> Thank you. We corrected this mistake with Figure 6. (p. 9, lines 287, 293-295)
- In page 6, line 182, S1_S2 should be S1+S2.
>> Thank you. We changed S1_S2 for S1+S2. (p. 6, line 183)
- A period is missing at the very end of the Discussion section (line 293)
>> Thank you. We added the period at the end of the Discussion (p. 10, line 318)
- Line 241, check for the correct use of parentheses.
>> We corrected the parentheses (p. 9, line 275)
- Importantly, the authors need to provide more of their opinions on the observed antibody levels, their differences and correlations among the cohorts, subclass, and target antigens. For example, why is the anti-S2 IgG levels are significantly different between infected vs. uninfected individuals, but not the anti-S1 antibodies or different subclasses (IgA and IgM)? What is the plausible explanation for the observed correlations shown in Fig. 5? Can any conclusions/predictions could be drawn from such hypotheses, especially regarding the protective effect of breast milk against COVID-19?
>> Thank you for this great comment. We added a paragraph to discuss the biological significance of milk antibodies against S2 subunit of SARS-CoV-2 in the discussion. (p. 8, line 227-236 and lines 259-263).
The plausible explanation for the observed correlations in Fig. 5 could be the polyreactive capacity to bind different epitopes for SARS-CoV-2-reactive SIgA/IgA and SIgM/IgM (Line 240).
We added in the conclusion that the mothers with a confirmed COVID-19 PCR or with previous viral symptoms may produce B cell response cross-reactive to S2 subunit that promote a broad protection against human coronaviruses and future SARS-CoV-2 mutations to their breastfed infants. The high numbers of positive correlations between antigens and secretory antibodies (SIgA and SIgM) in the COVID-19 PCR group and the absence of significant difference in SIgA or SIgM levels specific to S1 or S2 subunit SARS-CoV-2 between COVID-19 PCR and un-exposed mothers could be related to their polyreactive capacity to bind different epitopes on the subunits S1 and S2. The neutralizing capacity of antibodies between COVID-19 PCR, viral symptom, and pre-pandemic groups remains to be determined to draw a clear conclusion regarding the protective effect of human milk antibodies against SARS-CoV-2. S1+S2-reactive HCoV-OC43 IgG was higher in the COVID-19 group than in the Ctl1-2018 group but did not differ for HCoV-229E, revealing a stronger cross-reactivity between the S2 subunits of SARS-CoV-2 and human beta-coronaviruses than alpha-coronaviruses (p. 11, lines 393-406).
Round 2
Reviewer 1 Report
The manuscript can be accepted in present form